# Assessment of Environmental Risks of Particulate Matter Emissions from Road Transport Based on the Emission Inventory

Katarzyna Bebkiewicz [1] , Zdzisław Chłopek [2] , Hubert Sar [2,*] , Krystian Szczepański [3] and Magdalena Zimakowska-Laskowska [1]

1   National Centre for Emissions Management (KOBiZE), Institute of Environmental Protection—National Research Institute, 132/134 Chmielna Str., 00-805 Warsaw, Poland; katarzyna.bebkiewicz@kobize.pl (K.B.); magdalena.zimakowska-laskowska@kobize.pl (M.Z.-L.)

2   Institute of Vehicles and Construction Machinery Engineering, Warsaw University of Technology, 84 Narbutta Str., 02-524 Warsaw, Poland; zdzislaw.chlopek@pw.edu.pl

3   Institute of Environmental Protection—National Research Institute, 5/11D Krucza Str., 00-548 Warsaw, Poland; krystian.szczepanski@ios.edu.pl

*   Correspondence: hubert.sar@pw.edu.pl; Tel.: +48-22-234-8545

**Abstract:** The aim of this study is to investigate the environmental hazards posed by solid particles resulting from road transport. To achieve this, a methodology used to inventory pollutant emissions was used in accordance with the recommendations of the EMEP/EEA (European Monitoring and Evaluation Programme/European Economic Area). This paper classifies particulates derived from road transport with reference to their properties and sources of origin. The legal status of environmental protection against particulate matter is presented. The emissions of particulate matter with different properties from different road transport sources is examined based on the results of Poland's inventory of pollutant emissions in the year 2018. This study was performed using areas with characteristic traffic conditions: inside and outside cities, as well as on highways and expressways. The effects of vehicles were classified according to Euro emissions standards into the categories relating to the emissions of different particulate matter types. The results obtained showed that technological progress in the automobile sector has largely contributed to a reduction in particulate matter emissions associated with engine exhaust gases, and that this has had slight effect on particulate matter emissions associated with the tribological processes of vehicles. The conclusion formed is that it is advisable to undertake work towards the control and reduction of road transport particulate matter emissions associated with the sources other than engine exhaust gases.

**Keywords:** particulate matter; TSP; PM2.5; PM10; road transport; environmental protection

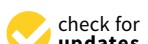



## 1. Introduction

### 1.1. Background Information on Particulate Matter Emissions from Road Transport

The sources of particulate matter (PM) emissions from road transport are [1–13]:

- engine exhaust gases;
- tribological processes of a motor vehicle that cooperates with its environment, specifically the particulates derived from the braking system, friction clutch, car parts subject to wear and tires, as well as the surfaces of roads.

The concentration of particulate matter in the air is increased by dust stirred up from the road's surface; this is the so-called secondary lift of dust, which has not been so far included in the inventory of pollutant emissions [5,14].

In addition, the structure and concentration of dusts in the atmosphere change as a result of environmental phenomena (e.g., absorbing various substances, mainly in a liquid or gaseous state). Modeling these phenomena is a very extensive problem that is beyond

the scope of a single study. Therefore, it was decided to use the standard produced by the pollutant emission inventory procedure, in accordance with the 2006 IPCC Guidelines for National Greenhouse Gas Inventories and the EEA/EMEP Emission Inventory Guidebook 2019 [8]. In this procedure, all aspects of the formation of solid particles in connection with the operation of motor vehicles are included in a systemic and contractual manner as a standard in the inventory of pollutant emissions.

Particulate matter is defined as a mixture of fine particles suspended in a gaseous dispersive phase such as air [5,6,14–18].

Particles are classified by their conventional dimensions, which depend on their average aerodynamic diameter. They are distinguished as [5–8,19–23]:

- total suspended particles (TSP)—with an average aerodynamic diameter of less than 300 μm;
- fine particles PM10—with an average aerodynamic diameter of less than 10 μm;
- fine particles PM2.5—fine inhalable particles with an average aerodynamic diameter less than 2.5 μm and colloidal fragmentation;
- ultrafine particles PM1—with an average aerodynamic diameter of less than 1 μm, practically invisible, distinguished in tests on internal combustion engines (particles with an average aerodynamic diameter of less than 100 nm, called nanoparticles, are separated from PM1; sometimes solid particles with dimensions smaller than 50 nm are also distinguished).

Particulate matter emissions from road transport are modeled depending on the characteristics of motor vehicles, including their intended use and conventional size, environmental quality of internal combustion engines as regards their emissions, and traffic conditions [6–8,10,11,21,23–25].

### 1.2. Particulate Matter Environmental Hazards and the Protection of Environment against Particulate Matter Emission from Road Transport—Legal Status

Particulate matter has adverse effects both on human health and the natural environment [5,9,14,26,27], including soil, water and living organisms. Once the air contains more water vapor, the aerosol's hygroscopic growth causes an increase in PM that results in poorer visibility and promotes the formation of fog and cold smog [28].

The most dangerous particles to human health are fine and ultrafine particles—PM2.5 and PM1 [5,9,12,14,26,27].

Particulate matter can enter the human body directly, through the respiratory tract, and indirectly, through the digestive system, with contaminated food or water (e.g., via heavy metals that are components of particulate matter). Particulate matter is soluble, both in water and in body fluids such as saliva, blood or gastric juice; therefore, it poses serious hazards to our health.

The impacts of particulate matter on humans are severe, and can even lead to death (e.g., due to cold smog), respiratory diseases (including chronic obstructive pulmonary disease (COPD)) and cardiovascular diseases [9,12,27].

The Constitution of the Republic of Poland (founded on 2 September 1997) articulates an obligation to protect the country's environment. The principal regulations of environmental law are set forth in the Environmental Protection Law of 27 April 2001. Relevant legal acts are also enforced that comply with legal regulations at the level of the European Union (EU).

Before 2008, four directives and one EC decision were enforced with regard to ambient air quality in the EU. On 21 May 2008, these regulations were consolidated into one policy document (i.e., Directive 2008/50/EC of the European Parliament and of the Council on ambient air quality and cleaner air for Europe), the so-called CAFE (Clean Air For Europe) Directive. Table 1 outlines the provisions of five legal acts [5,8,14,29]:

**Table 1.** Provisions of the legal acts [5,8,14,29].

| Identification Data of the Legal Act | Issue Covered by the Legal Act |
|---|---|
| Council Directive 96/62/EC of 27 September 1996 | Ambient air quality assessment and management. |
| Council Directive 1999/30/EC of 22 April 1999 | Limit values for sulphur dioxide, nitrogen dioxide and oxides of nitrogen, particulate matter and lead in ambient air. |
| Directive 2000/69/EC of the European Parliament and of the Council of 16 November 2000 | Limit values for benzene and carbon monoxide in ambient air. |
| Directive 2002/3/EC of the European Parliament and of the Council of 12 February 2002 | Ozone in ambient air. |
| Council Decision of 27 January 1997 | Establish a reciprocal exchange of information and data from networks and individual stations measuring ambient air pollution within Member States. |

As part of air quality monitoring, monitoring stations measure concentrations of:

- PM10 particulate matter;
- PM2.5 particulate matter;
- PM1 particulate matter.

Table 2 outlines the documents that, among others, are currently enforced by the European Union in regards to vehicular emissions at the level of conventionally known Euro 6/VI regulations.

**Table 2.** Documents currently enforced by the European Union at the level of conventionally known Euro 6/VI regulations.

| Identification Data of the Legal Act | Issue Covered by the Legal Act |
|---|---|
| Commission Regulation (EC) No 692/2008 of 18 July 2008 | Implementing and amending Regulation (EC) No 715/2007 of the European Parliament and of the Council on type-approval of motor vehicles with respect to emissions from light passenger and commercial vehicles (Euro 5 and Euro 6) and on access to vehicle repair and maintenance information. |
| Commission Regulation (EU) No 566/2011 of 8 June 2011 | Amending Regulation (EC) No 715/2007 of the European Parliament and of the Council and Commission Regulation (EC) No 692/2008 as regards access to vehicle repair and maintenance information. |
| Regulation (EC) No 595/2009 of the European Parliament and of the Council of 18 June 2009 | Type-approval of motor vehicles and engines with respect to emissions from heavy duty vehicles (Euro VI) and on access to vehicle repair and maintenance information and amending Regulation (EC) No 715/2007 and Directive 2007/46/EC and repealing Directives 80/1269/EEC, 2005/55/EC and 2005/78/EC. |
| Commission Regulation (EU) No 582/2011 of 25 May 2011 | Implementing and amending Regulation (EC) No 595/2009 of the European Parliament and of the Council with respect to emissions from heavy duty vehicles (Euro VI) and amending Annexes I and III to Directive 2007/46/EC of the European Parliament and of the Council. |
| Commission Regulation (EU) No 64/2012 of 23 January 2012 | Amending Regulation (EU) No 582/2011 implementing and amending Regulation (EC) No 595/2009 of the European Parliament and of the Council with respect to emissions from heavy duty vehicles (Euro VI) |

**Table 2.** *Cont.*

| Identification Data of the Legal Act | Issue Covered by the Legal Act |
|---|---|
| Commission Regulation (EC) 715/2007 of the European Parliament and of the Council of 20 June 2007 | Type approval of motor vehicles with respect to emissions from light passenger and commercial vehicles (Euro 5 and Euro 6) and on access to vehicle repair and maintenance in-formation, European Commission (EC), Official J. European Union, L 171, 2007. |
| Commission Regulation (EU) 2016/427 of 10 March 2016 | Amending Regulation (EC) No. 692/2008 as regards emissions from light passenger and commercial vehicles (Euro 6), Verifying Real Driving Emissions, Official J. European Union, L 82, 2016. |
| Commission Regulation (EU) 2016/646 of 20 April 2016 | Amending Regulation (EC) No. 692/2008 as regards emissions from light passenger and commercial vehicles (Euro 6), Verifying Real Driving Emissions, Official J. European Union, L 109, 2016. |
| European Commission (2017) Regulation (EC) 2017/1151 of 1 June 2017 | Supplementing Regulation (EC) No 715/2007 of the European Parliament and of the Council on type-approval of motor vehicles with respect to emissions from light passenger and commercial vehicles (Euro 5 and Euro 6) and on access to vehicle repair and maintenance information, amending Directive 2007/46/EC of the European Parliament and of the Council, Commission Regulation (EC) No 692/2008 and Commission Regulation (EU) No 1230/2012 and repealing Commission Regulation (EC) No 692/2008. Official Journal of the European Union. L 175. |

The aforementioned regulations comprise general and specific requirements for the testing of motor vehicles and internal combustion engines as regards their emissions. In the case of particulate matter, the object of testing is the emission (mass) of particulates and the number of particulates. The physical quantities described below are assumed as criterion quantities [30,31].

For vehicles with a reference mass of less than 2.61 Mg (passenger cars, light trucks and minibuses), the criterion quantities are as follows:

- The specific distance particulate matter emission ($b_{PM}$)

The specific distance particulate matter emission is the derivative of particulate matter emission ($m_{PM}$) relative to the distance(s) travelled by the vehicle.

$$b_{PM} = \frac{m_{PM}}{ds}, \tag{1}$$

- The specific distance number of particulate matter ($b_{PN}$):

The specific distance number of particulate matter is the derivative of the particulate matter (PN) relative to the distance(s) travelled by the vehicle.

$$b_{PN} = \frac{PN}{ds}, \tag{2}$$

For internal combustion engines of vehicles with a reference mass greater than 2.61 Mg (trucks and buses):

- The specific brake emission of particulate matter ($e_{PM}$)

The specific brake emission of particulate matter is the derivative of the emission of particulate matter ($m_{PM}$) relative to the internal combustion engine work (L).

$$e_{PM} = \frac{m_{PM}}{dL}, \tag{3}$$

- The specific brake number of particulate matter ($e_{PN}$)

The specific brake number of particulate matter is the derivative of the particulate matter number (PN) relative to internal combustion engine work (L).

$$e_{PN} = \frac{PN}{dL},$$ (4)

Figure 1 presents the limits of the specific brake emissions of particulate matter for stages Euro V and Euro VI tested by static WHSC (World Harmonized Stationary Cycle) and in the dynamic tests ETC (European Transient Cycle) for Euro V and WHTC (World Harmonized Transient Cycle) for Euro VI [30,31]. The specific brake number of particulate matter is limited from stage Euro VI onwards.

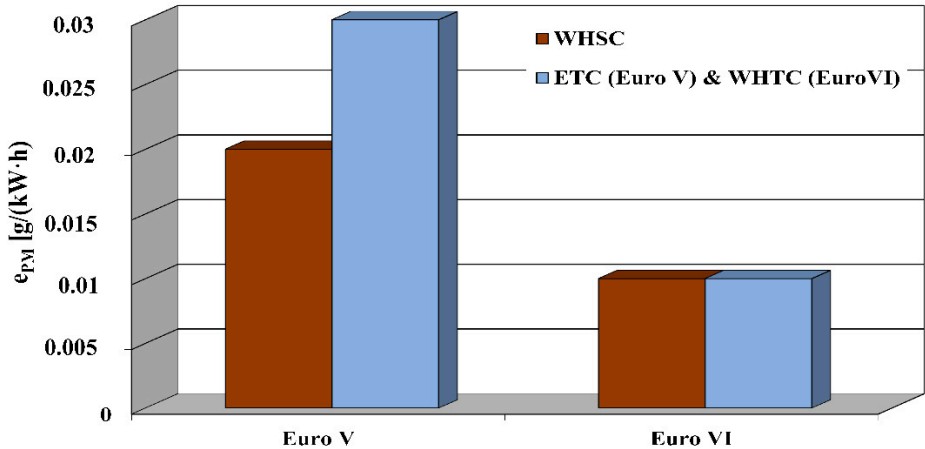

**Figure 1.** Limits of the specific brake emission of particulate matter using the static test WHSC (World Harmonized Stationary Cycle) for Euro V and Euro VI emission stages, and the dynamic tests ETC (European Transient Cycle) and WHTC (World Harmonized Transient Cycle) for Euro V and Euro VI, respectively (consistent with [30]).

The progress is clearly visible in the protection of the environment against particulate matter emitted from self-ignition engines for heavy-duty vehicles—a twofold reduction in the limits for the emissions of particulate matter in the static test and a threefold reduction in the dynamic test.

This progress results from the dynamic development of technical solutions [32,33], including use of high fuel injection pressure in common-rail injection systems, which control air turbulence in the combustion chamber. First of all, there has been progress in the field of exhaust gas cleaning—mainly self-regenerating, catalytic diesel particulate filters, which are a component of integrated exhaust gas treatment systems. The development of engine fuels is also important, including the use of engine oils with a very low sulphur content.

### 1.3. Current State of Knowledge on Particulate Matter Emissions from Road Transport

There have been numerous papers published on particulate matter emissions from engine exhaust. Publications in the field of particulate matter emissions from internal combustion engines concern both the research methodology and analysis of empirical test results, as well as the results of modeling particulate matter emissions. In [34], the procedures and methodologies of the UN/ECE PMP Working Group are presented in order to evaluate their abilities to test particulate emissions and numbers. Results of inter-laboratory tests of particulate matter emissions are presented for various types of internal combustion engines, with particular emphasis on the repeatability and reproducibility of measurement data.

The modeling of particulate matter emissions from internal combustion engines is the subject of [35–37]. In [35], the available models of particulate matter emissions

from exhaust systems are analyzed, mainly in the form of zero-dimensional emission characteristics comprising various types of emission factors. These are most often specific distance emissions of particulate matters and specific brake emissions of particulate matters, characteristics that enable the modeling of particulate matter emissions from exhaust systems depending on, for example, the nature of vehicle traffic. This article proposes an original model of the emission of particulate matters from vehicle engines, and presents the results of simulation tests of particulate matter emissions.

In [36], the subject of pollutant emission inventory is studied in the context of, inter alia, testing of particulate emissions from exhaust systems of the internal combustion engines of motor vehicles. The article presents considerations of the inventory of particulate matter emissions and air quality modeling in the Pearl River Delta region—a region with particularly high environmental pollution. The main purpose of the work is to assess air quality modeling methods in order to simulate pollutant concentrations. Three modeling models were assessed under conditions typical of winter and summer periods. The best results of compliance for the concentration measurements were obtained for the EDGAR (Emission Database for Global Atmospheric Research) model [37].

Most of the works concern the analysis of the results of empirical tests of particulate matter emissions from combustion engines.

Ref. [38] summarizes the evolution of particulate matter emissions from spark ignition engines marketed from the early 1990s to 2019 in different regions of the world. Significant progress has been made in this regard, mainly due to the technical progress of internal combustion engines. This applies not only to the improvement of fuel dosing and combustion systems, but above all to exhaust gas cleaning systems. The significant effect of ambient temperature on particulate matter emissions is highlighted.

Specific exhaust gas cleaning systems are the subject of [31]. This paper presents the results from testing a particulate matter filter. The tests were mainly carried out under the conditions of a low engine load typical of passenger cars in cities.

Ref. [39] discusses the emission of various quantities of particulate matter from compression-ignition and spark-ignition engines. The authors point out that more precursors of very harmful organic aerosol are produced as a result of petrol combustion in a spark-ignition engine. In [40], it is shown that gasoline-powered spark ignition engines emit more particulate matter compared to compression ignition engines equipped with particle filters (DPF).

Much less information is available on road transport particulate matter emissions from sources other than engine exhaust.

The models of particulate matter emissions related to road transport that have been developed [14,21,22,30] are among the models in accordance with the criterion of functional similarity (the so-called behavioral models) [41]. These models make the total particulate matter emission from both the internal combustion engine and the vehicle tribological sources dependent on properties such as driving speed, vehicle type and weight and road surface.

Refs. [22,42] present a literature review on various aspects of particulate matter emissions from sources other than exhaust, in particular from brake and tire wear.

Refs. [10,12,13] describe an original system for reducing the emissions of particulate matters from braking systems, analyzing the dimensions and mineral and chemical composition of particulate matter generated in braking systems. The developed system enabled a reduction of particulate matter emissions from braking systems by approximately 90%. The studies found that particulate matters from braking systems are particularly dangerous to health due to their compositions and dimensions. The material of these particles are heavy metals and their compounds mainly consist of iron. The dimensions of these particles are predominantly less than 1 μm.

Ref. [42] concerns the emission of particulate matters from braking systems. The authors of the publication proposed an original research methodology for the analysis of particulate matter emissions and determine the volume of particulate matter in a car's

braking system. This methodology consisted of applying the WLTP (Worldwide Harmonized Light Vehicle Test Procedure) test [32,33] to a chassis dynamometer (i.e., under the conditions of type approval tests). Under these comparable driving conditions, the authors investigated the influence of numerous operating factors on the emissions of particulate matters from the braking system.

Ref. [43] highlights the problem of information availability with respect to the properties and qualities of road transport particulates from sources other than fuel combustion. They also investigate the limited information available on particulate emissions from abrasion.

Ref. [26] systematically reviews and analyzes the available results of studies on particulate matter (PM10 and PM2.5) source apportionments carried out in cities to estimate the typical source contributions of particulate matter in different countries and regions. Based on the available information, 25% of PM2.5 emissions in cities worldwide come from traffic, 15% from industrial activities, 20% from domestic combustion, 22% from unspecified sources of anthropogenic origin, and 18% from natural sources (biological and mineral particles).

Ref. [44] presents the results of a study on particulate matter emissions from road transport. Most of the particles were composites of tire, road surface and brake system materials. Particulate matter varied in size distribution, composition and structure, depending on travel velocity, traffic volume and rolling stock.

Ref. [18] states that particulate matter emissions from other traffic-related sources become an increasingly important issue as emissions of exhaust pollutants decrease. Among others, particulate matter emissions from tire and road surface wear may be responsible for 5–10% of microplastics deposited in the oceans.

In [45], the authors indicate that statistical studies confirm a smaller share of PM10 emissions from mobile sources in cities when compared with stationary sources (53%).

Ref. [39] presents the results of a study on PM2.5 emissions from tire and road surface abrasion in London, Tokyo and Los Angeles. It was found that the proportion of PM2.5 particles from tire and pavement abrasion was only 0.27% of the total PM2.5 concentration.

Based on the official results of emission inventories, few publications have discussed the emission of particulate matter from road transport. The present study concerns an assessment of the effects of the motor vehicles category under specific traffic conditions on the emissions of different particulate matters.

Therefore, the aim of this study is to investigate the environmental risks of particulate matter from road transport on the basis of the official results of the pollutant emission inventory, in accordance with the methodology recommended by EMEP/EEA (European Monitoring and Evaluation Programme/European Economic Area) [8]. A recommended tool in the form of the COPERT 5 software was used to determine the national annual emission of dust from road transport [7]. The authors decided to verify the research hypothesis in regards to the significant impact of traffic inside and outside cities, as well as on highways and expressways, on particulate matter emissions. The second research hypothesis concerns a significant share in the emission of particulate matters from sources other than the exhaust systems of internal combustion engines.

## 2. Methodology for Studying Environmental Risks of Particulate Matter from Road Transport Based on Emission Inventory

The study of environmental hazards associated with the emission of road transport particulate matter was carried out on the basis of the results of the official inventory of pollutant emissions from road transport in Poland in 2018, prepared by the National Centre for Emissions Management (KOBIZE), the Institute of Environmental Protection—National Research Institute [29].

The inventory of emissions from road transport was carried out in compliance with the European Monitoring and Evaluation Programme (EMEP)/European Environmental Agency (EEA) methodology [8] with the use of COPERT 5 software [7].

Motor vehicles are classified into categories. In a philosophical sense, a category (comes from the Greek κατηγορείν—kategorein—to adjudge) is a concept introducing a structure: a class of objects having certain interrelated characteristics.

The motor vehicle categories are divided according to various criteria, primarily [1–4,7,8,11,24,29]:

- motor vehicle purpose of use;
- motor vehicle conventional size and propulsion engine;
- characteristics of the motor vehicle and its propulsion engines as regards, among others, the engine cycle, detailed technical solutions and performance, primarily with respect to pollutant emissions;
- power fuel for the motor vehicle's internal combustion engine;
- technical level of motor vehicle and its propulsion engine.

The elementary category of road vehicles [1–4,24] comprises motor vehicles that all share the same essential characteristics (e.g., passenger cars with a compression-ignition engine with a displacement of more than 2 dm$^3$, Euro VI stage, powered by diesel fuel).

The cumulative category of road vehicles [1–4,24] comprises motor vehicles that do not all share the same essential characteristics (e.g., passenger cars with spark ignition engines).

The most cumulative motor vehicle category includes all motor vehicles.

According to the procedure used in the inventory of pollutant emissions from road transport, national annual emissions of particular pollutants are determined for the elementary categories of motor vehicles.

The subject of research presented in this study is the emissions recorded for the cumulative categories of motor vehicles, including [1–4,7,8,11,24,29]:

- passenger cars;
- light commercial vehicles;
- heavy duty trucks;
- buses;
- mopeds and motorcycles (and microcars and quads).

An example of differentiating motor vehicles according to detailed characteristics is the division of the category of buses into city buses and coaches, and the category of trucks into rigid and articulated.

Due to the cycles of the internal combustion engines, engines are divided into two-stroke and four-stroke.

The conventional size of a vehicle is characterized according to [1–4,7,8,11,24,29]:

- internal combustion engine displacement—for passenger cars, light trucks, motorcycles and mopeds (and possibly quads);
- maximum mass—for trucks and buses and, additionally, light trucks.
- Vehicles are distinguished according to their fuel source, including the following [1–4,7,8,11,24,29]:
- motor gasoline;
- diesel fuel;
- fuels based on organic oil esters (so-called biodiesel);
- ethyl alcohol;
- liquefied petroleum gas (LPG);
- natural gas.

The criterion for the division of motor vehicles according to pollutant emissions is dependent on meeting pollutant emission requirements in compliance with the effective regulations on European emission standards [1–4,7,8,24,29–31].

Specific technical solutions include hybrid and plug-in hybrid car categories.

The inventory of pollutant emissions is conducted for substances that are harmful to living organisms and the environment. Table 3 presents different ways of classifying particulate matters and their emission sources.

**Table 3.** Different classifications of particulate matters and their emission sources.

| Sources of Particulate Matters or Their Emission Sources Classification | Classification of Particulate Matters or Their Emission Sources |
| --- | --- |
| Classification of particulate matters for the needs of the inventory [7,8,29] | total suspended particles (TSP); fine particles PM10 (PM10 particulate matter); fine particles PM2.5 (PM2.5 particulate matter); soot (black carbon) |
| Particulate matter emissions, TSP, PM10 and PM2.5 distinguished by [7,8,29] | total emissions; emissions from sources other than engine exhaust |
| Sources of particulate matter emissions other than engine exhaust including PM emission from vehicle tribological systems interacting with the environment, mainly [5–8,21,24,29] | from vehicle braking systems; from vehicle tires and roadway surfaces |

Vehicle traffic characteristics are determined for model traffic conditions, such as [1–4,7,8,24,29]:

- inside cities;
- outside cities;
- on motorways and expressways (highways).

For each traffic model, certain values are assumed, including [1–4,7,8,24,29]:

- vehicle average velocity;
- share of the distance travelled by vehicles in a given traffic model versus the total distance travelled in all traffic models.

The extensive quantities for each elemental category in the emissions inventory are [1–4,7,8,24,29]:

- number of motor vehicles;
- average annual mileage of motor vehicles.

Information on total consumption of each fuel type constitute additional statistics for the inventory. This information is used for scaling vehicle mileage data to determine fuel consumption results based on emission modeling and statistical data.

The characteristics of pollutant emissions from road transport in the COPERT model were determined from empirical research and pollutant emission models.

Particulate matter emissions were determined from the exhaust systems of internal combustion engines using tests of individual elementary categories focusing on motor vehicle driving tests carried out using a chassis dynamometer. These tests simulate various vehicle traffic conditions characterized primarily by the average speeds of vehicles. The results of the empirical tests make it possible to determine the dependence of specific distance emissions of the dimensional fractions of particulate matters on the average speeds of a vehicle.

In the case of dust emissions from tribological systems, emission models were used (e.g., the Lohmeyer model [23] and the EPA model (United States Environmental Protection Agency) [17]). These models were created from the results of empirical studies into the wear of tribological elements of vehicles during their operation [4,12,13].

Pollutant emission inventories are conducted annually. The determined emission of each pollutant in a given year is referred to as the national annual emission of a given pollutant.

The studies whose results are presented in the present paper were conducted for the cumulative categories: passenger cars, light trucks, city buses, long-distance buses, motorcycles and mopeds.

The official data used by KOBiZE for the inventory of pollutant emissions from road transport in Poland [29] were used in the present study. The cumulative category data adopted in the study are shown in Figures 1–4.

Figure 2 shows the numbers of vehicles in the cumulative categories that are the focus of this study.

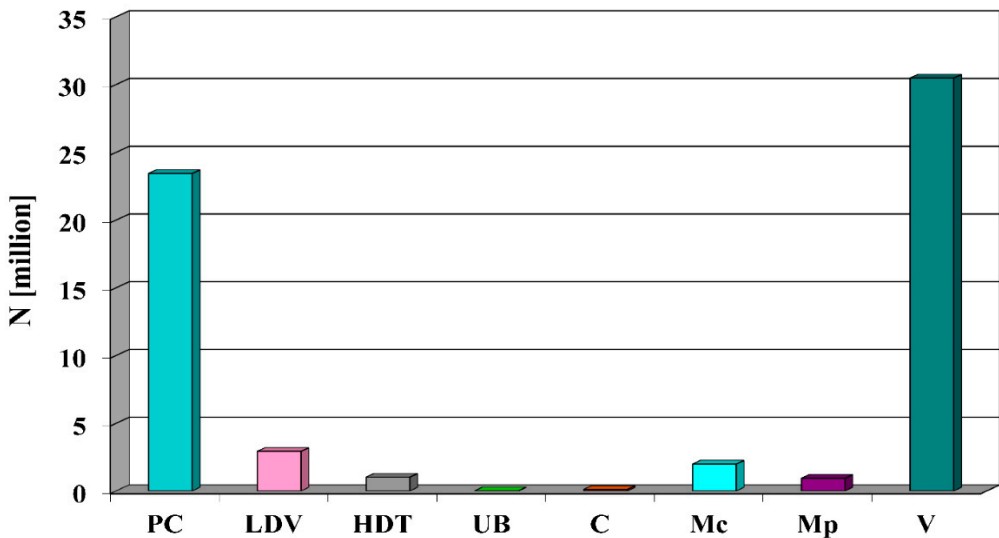

**Figure 2.** Number (N) of passenger cars (PC), light duty vehicles (LDV), heavy duty vehicles (HDT), urban buses (UB), coaches (C), motorcycles and quads (Mc), mopeds (Mp) and all vehicles (V).

Figure 3 shows the average annual mileage of vehicles in each cumulative category.

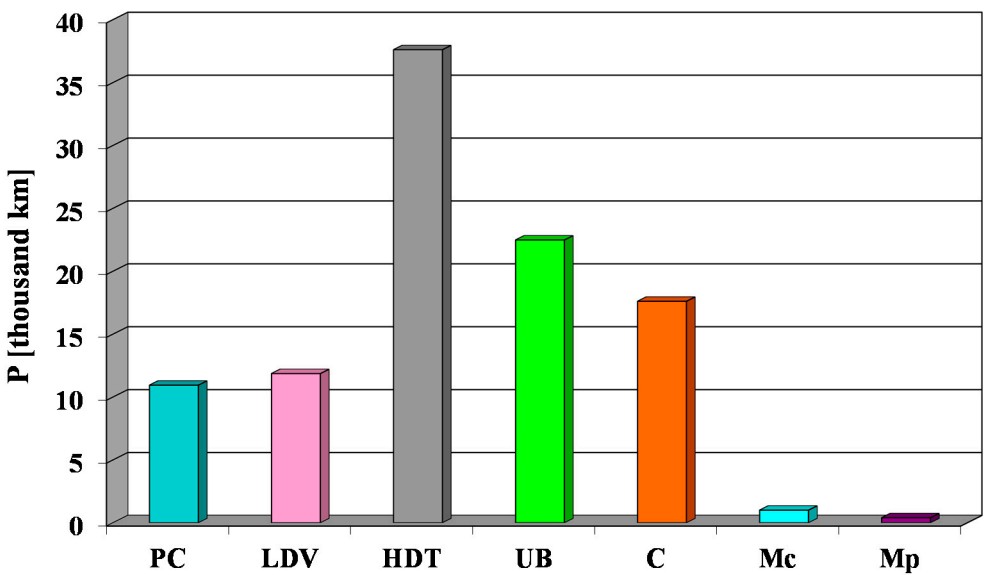

**Figure 3.** Annual mileage (P) of passenger cars (PC), light duty vehicles (LDV), heavy duty vehicles (HDT), urban buses (UB), coaches (C), motorcycles and quads (Mc) and mopeds (Mp).

Figure 4 shows the average travel velocity of each cumulative category of vehicles under traffic conditions: urban (inside cities), rural (outside cities) and on highways.
Figure 5 shows the share of the distance travelled by cumulative category vehicles under urban, rural and highway traffic conditions in the total distance travelled by these vehicles.

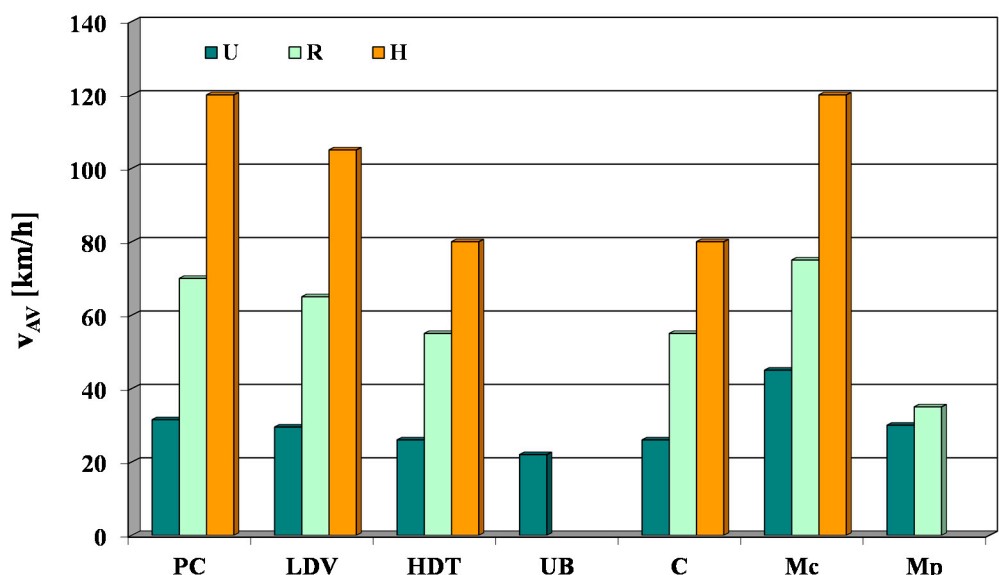

**Figure 4.** Average velocity ($v_{AV}$) of passenger cars (PC), light duty vehicles (LDV), heavy duty vehicles (HDT), urban buses (UB), coaches (C), motorcycles and quads (Mc) and mopeds (Mp) under different traffic conditions: urban (U), rural (R) and highway (H).

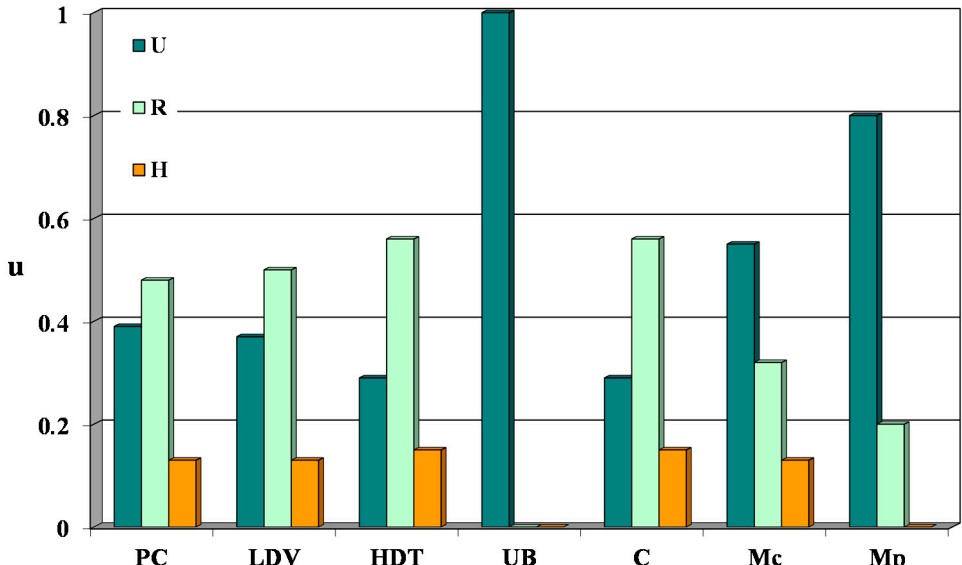

**Figure 5.** Share of distance (u) in urban (U), rural (R) and highway (H) traffic conditions at all distances for passenger cars (PC), light duty vehicles (LDV), heavy duty vehicles (HDT), urban buses (UB), coaches (C), motorcycles and quads (Mc) and mopeds (Mp).

The present study addresses the following key assessment criteria:

- share of particle emissions in total particulate matter emissions;
- share of particle emissions and total particulate matter emissions from sources other than vehicle exhaust in particle emissions in total particulate matter emissions from all sources;
- influence of vehicle traffic models on specific fractions of particulate matter emissions and on total emissions of particulate matter inside and outside cities, as well as on highways.

### 3. Results on Environmental Hazards from Road Transport Particulate Matter Based on the Emission Inventory

Figures 6–15 show an analysis of the particulate matter emissions from road transport.

Figure 6 shows the national annual emissions of the studied fractions of particulate matter from road transport.

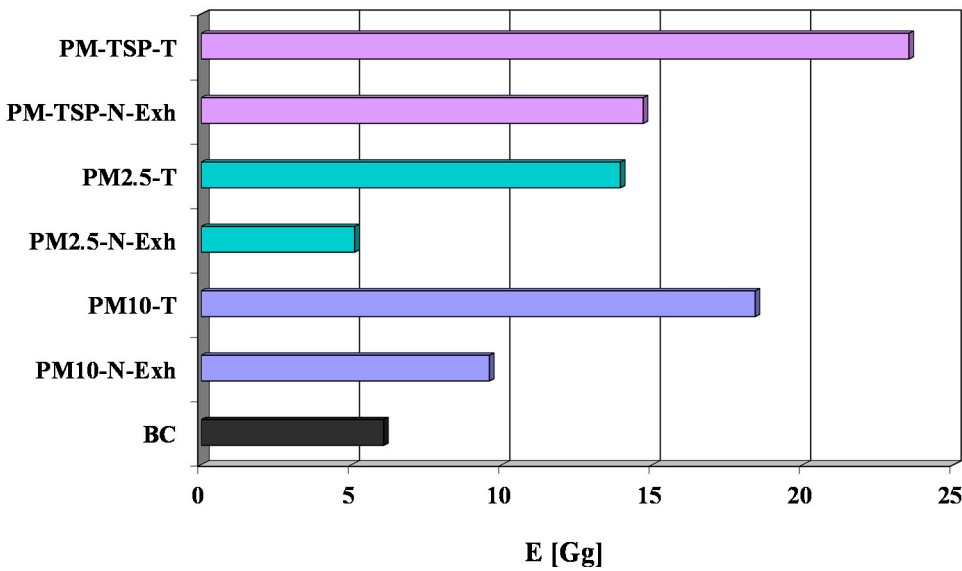

**Figure 6.** National annual emissions (E) of particulate matter from road transport.

The total particulate matter emissions from sources other than combustion engines was 62% of the total emissions. This indicates a high risk to the environment posed by particulate matter emissions from sources other than combustion engines, whereas technological developments in combustion engines have made it so their exhaust gases are no longer the main source of particulate matter leading to environmental harm.

The highest relative annual emissions from the fractions examined in this study was for PM10 particles (78%), whereas for PM10 particles from sources other than exhaust gases this was 41%. The smallest was the relative annual emission of PM2.5 from sources other than exhaust gases (22%).

Figure 7 shows the national annual emissions of total particulate matter from road transport as categorized by vehicle traffic pattern: urban, non-urban (rural), highway and all traffic conditions.

The national annual emissions of total particulate matter from traffic outside cities (45%) showed a similar value to that assessed for traffic in cities (44%). The lowest value of particulate matter emissions (12%) was observed in the case of highway traffic. Considerably high particulate matter emissions were observed in cities, which suggests a serious threat to health and the lives of many people, as these emissions affect highly populated urban areas.

Figure 8 shows the national annual emissions of particulate matter from sources other than engine exhaust as a function of vehicle traffic patterns.

The highest relative national annual emissions of particulate matter from sources other than engine exhaust gases were from vehicle traffic outside cities (nearly 50%), while in cities it was about 40% and on highways it was 10%.

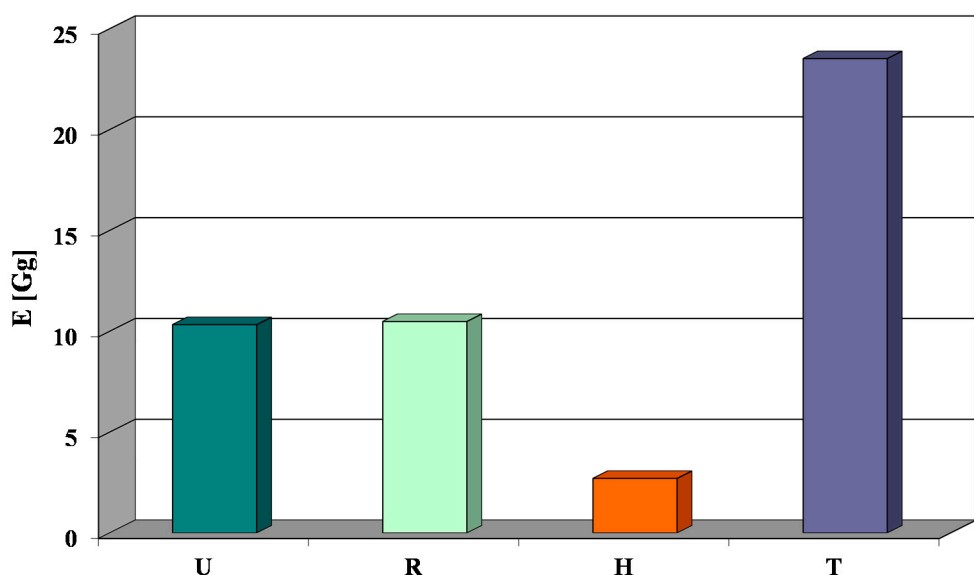

**Figure 7.** National annual emissions (E) of total suspended particles (TSP-T) from road transport under different traffic conditions: urban (U), rural (R), highway (H) and total (T).

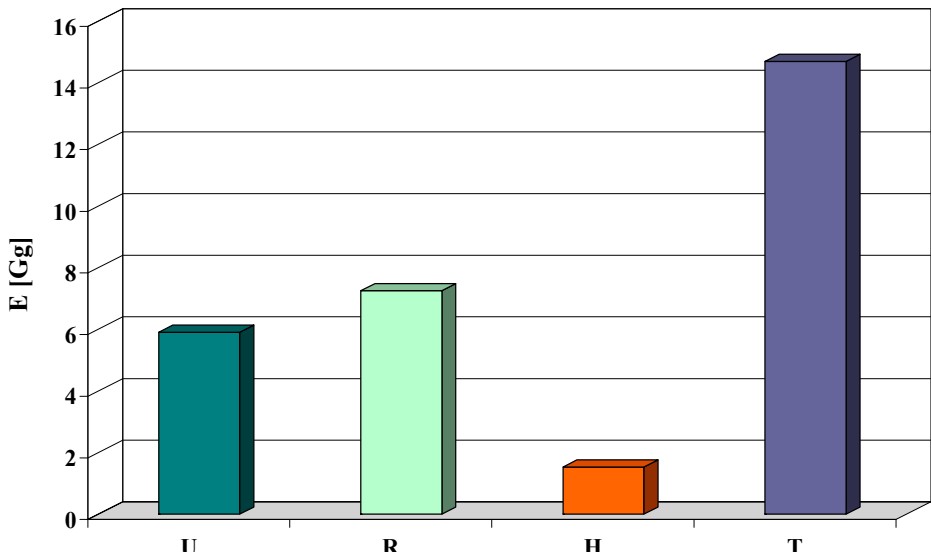

**Figure 8.** National annual emissions (E) of non-exhaust total suspended particles (PM-TSP-N-Exh) under different traffic conditions: urban (U), rural (R), highway (H) and total (T).

Figure 9 shows the national annual PM10 emissions as a function of vehicle traffic patterns.

The highest relative national annual emission of PM10 was observed for vehicle traffic in cities (46%), while it was slightly lower outside cities (43%) and lowest on highways (11%). Once again, the health of residents in urban areas is threatened, as they are highly exposed to adverse effects of fine particles (PM10).

Figure 10 shows the national annual emissions of PM10 from sources other than engine exhaust as a function of vehicle traffic patterns.

The highest relative national annual emissions of PM10 from sources other than engine exhaust were observed for vehicle traffic outside cities (nearly 50%), whereas in cities these emissions constituted 42% and on highways they made up less than 10%.

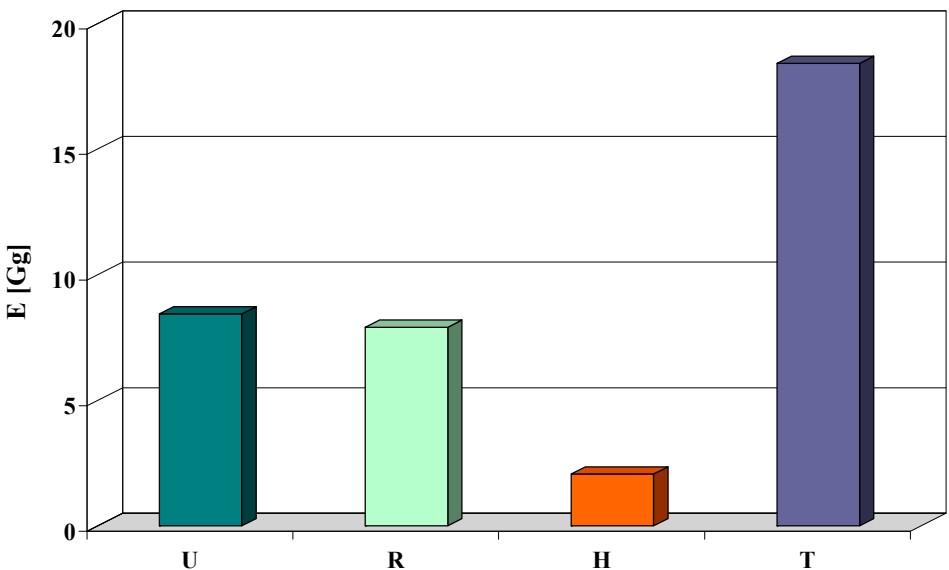

**Figure 9.** National annual emissions (E) of particulate matter PM10 (PM10-T) under different traffic conditions: urban (U), rural (R), highway (H) and total (T).

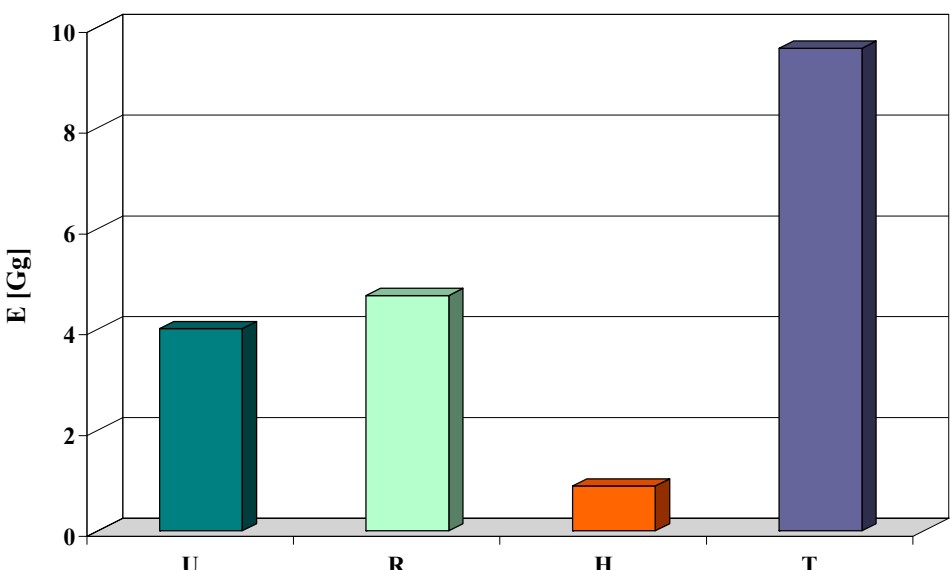

**Figure 10.** National annual emissions (E) of non-exhaust particulate matter PM10 (PM10-N-Exh) under different traffic conditions: urban (U), rural (R), highway (H) and total (T).

Figure 11 shows the national annual emissions of PM2.5 depending on vehicle traffic patterns.

From the research conducted it can be seen that the results were similar between PM2.5 and PM10 particles. The highest relative national annual emission of PM2.5 was observed in the case of traffic in cities (47%), while in traffic outside cities it was 41% and on highways it was only 12%.

Figure 12 shows the national annual emissions of PM2.5 from sources other than engine exhaust gases depending on vehicle traffic patterns.

From the conducted tests it can be seen that the relative results were similar between PM2.5 and PM10 particulate matter. The highest relative national annual emission of PM2.5 was for vehicle traffic outside cities (almost 50%), whereas in cities it was about 41% and on highways it was approximately 10%.

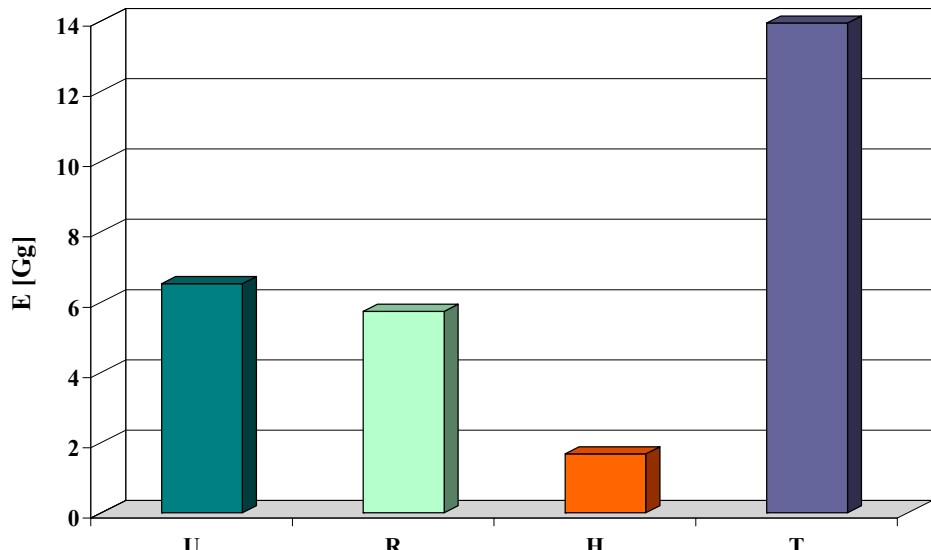

**Figure 11.** National annual emissions (E) of particulate matter PM2.5 (PM2.5-T) under different traffic conditions: urban (U), highway (H) and total (T).

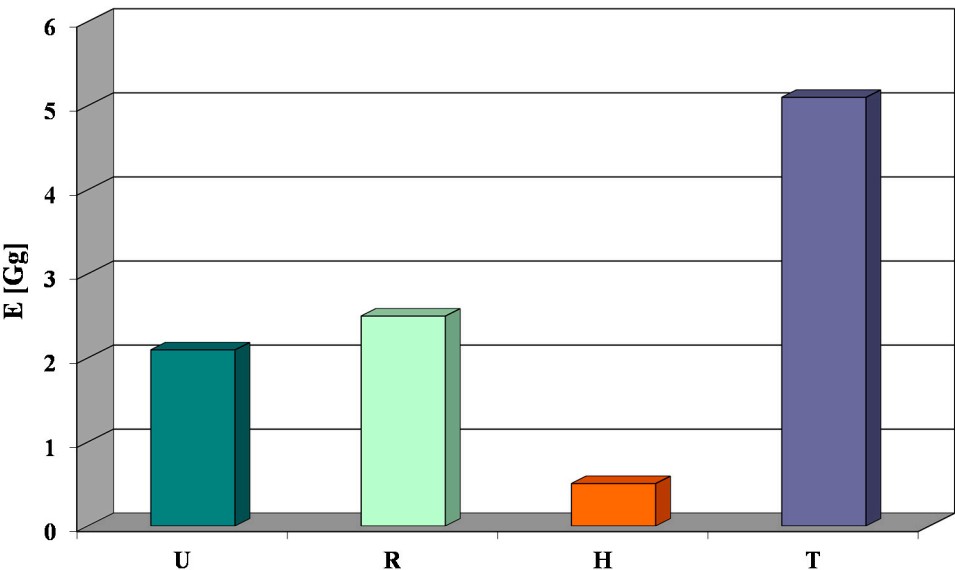

**Figure 12.** National annual emissions (E) of non-exhaust particulate matter PM2.5 (PM2.5-N-Exh) under different traffic conditions: urban (U), rural (R), highway (H) and total (T).

Figure 13 shows the national annual soot (black carbon) emissions as a function of vehicle traffic patterns.

The relative national annual soot emissions were highest in the case of urban traffic (51%), followed by non-urban traffic (36%), and at the lowest on highways (14%). Soot emissions are assumed to come only from engine exhaust.

Figures 14 and 15 summarize the results of the study: the national annual emissions of particulate matter (Figure 14) and the relative national annual emissions of particulate matter (Figure 15) from road transport, depending on the vehicle traffic model.

The average values of the relative national annual emissions of all particulate matter were similar for urban and non-urban traffic conditions (i.e., almost 44%). For traffic on highways, the average value of the relative national annual emissions of all particulate matter was about 12%.

The obtained research results are not only cognitive, but also practical. The practical nature of the research results consists primarily of indicating which traffic and vehicle categories contribute to the greatest threats to human health and the environment from dust. A better of understanding of this phenomenon enables a reshaping of the impacts of vehicle traffic not only through technological but also legal means.

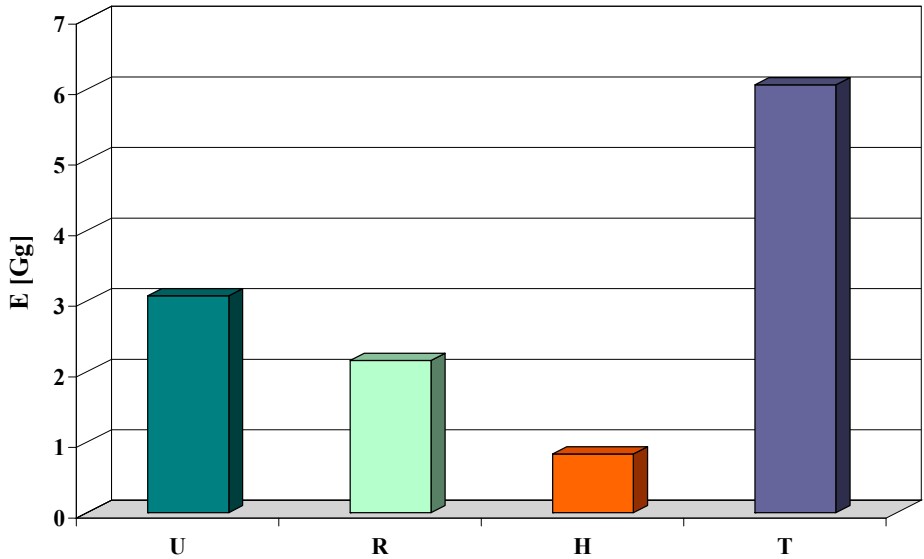

**Figure 13.** National annual emission (E) of black carbon (BC) under different traffic conditions: urban (U), rural (R), highway (H) and total (T).

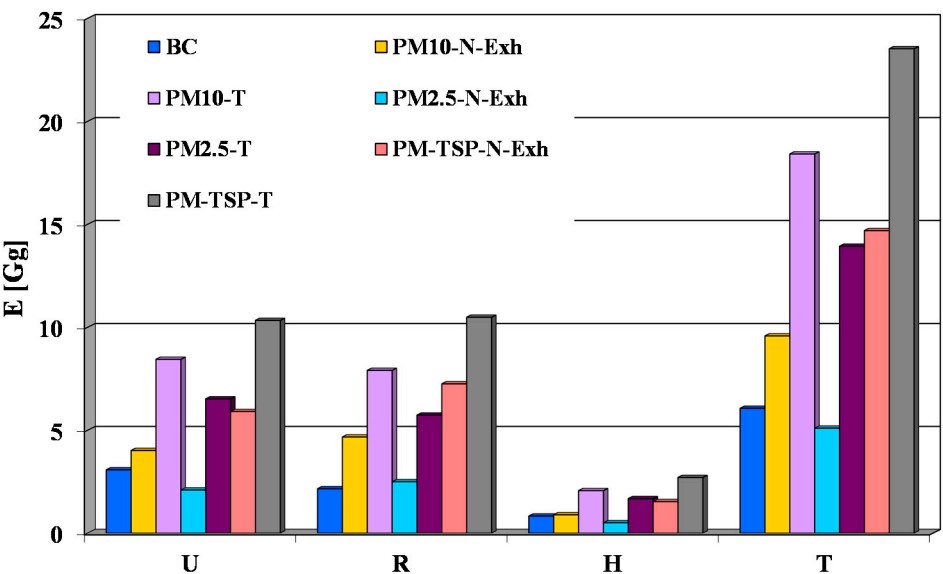

**Figure 14.** National annual emissions (E) of particulate matter from road transport under different traffic conditions: urban (U), rural (R), highway (H) and total (T).

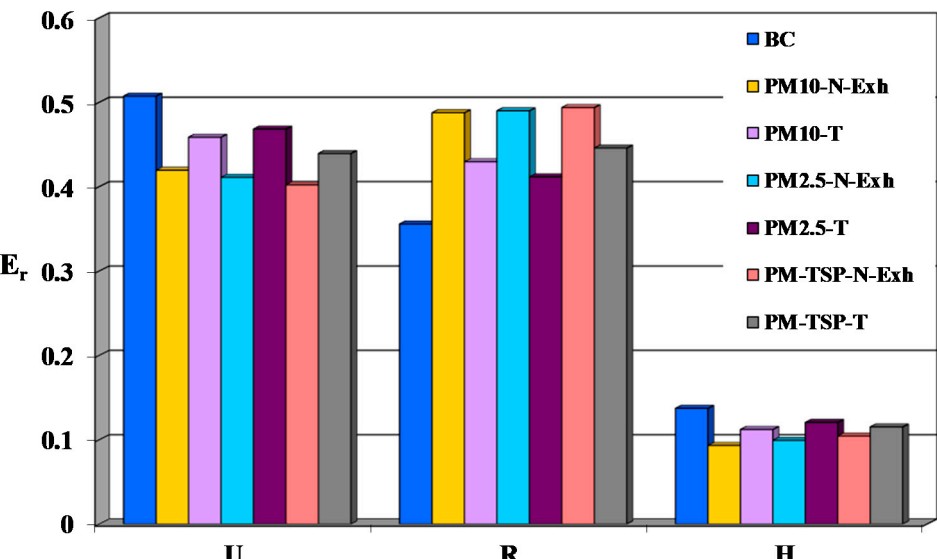

**Figure 15.** Relative national annual emissions (Er) of particulate matter from road transport under different traffic conditions: urban (U), rural (R), highway (H) and total (T).

### 4. Conclusions

The present study was carried out based on data derived from the inventory of pollutant emissions from road transport in Poland in 2018. Essentially, this study aimed to assess the makeup of particulate matter emissions according to the characteristics of motor vehicles, such as vehicle purpose and conventional size; ecological quality of internal combustion engine in terms of pollutant emissions; and traffic conditions, defined as traffic inside and outside cities and on highways.

The conducted research confirms the research hypotheses considered in the introduction. It is confirmed that motor vehicle traffic has a significant impact on particulate emissions. A significant share of particulate matter emissions from sources other than the exhaust systems of internal combustion engines was also confirmed.

Based on the study results, the following conclusions can be drawn:

1.  The highest relative annual emission of PM10 particles (78%) and PM10 from sources other than engine exhaust gases was 41%. The relative annual emission of PM2.5 from sources other than exhaust gases was the lowest (22%).
2.  The total particulate matter emissions from sources other than exhaust gases constituted 62% of PM total emissions.
3.  The highest relative national annual emissions of particulate matter from sources other than engine exhaust gases were observed for urban and non-urban vehicle traffic (40–50%), and the lowest was observed on highways (10%).

The most significant finding is that the main source of particulate matter emission was not engine exhaust gases but other sources, foremost vehicle braking systems [5,14,21,33] followed by the contact between vehicle tires and road surfaces [5]. A substantial progress in the construction of internal combustion engines and the development of methods for reducing pollutant emissions from these engines have largely contributed to a considerable decrease in pollutant emissions from motor vehicles. This is evident in the limitations on exhaust emissions that have been enforced for motor vehicles and their engines, as well as for light vehicles (passenger cars, light trucks, microbuses, motorcycles and mopeds) and heavy vehicles (trucks and buses) [30,31].

Particulate matter emissions from vehicle braking systems consist of very small particles with an equivalent diameter of less than 1 μm. Additionally, these emissions contain mainly heavy metals in their compounds, especially iron [14,21,33]. On the other hand, particulate matter produced during contact between vehicle tires and the surface of a road is

characterized by particles of larger dimensions containing very harmful substances, including organic compounds such as polycyclic aromatic hydrocarbons and their derivatives [5] (the substances that promote the formation of cancer).

The risk associated with particulate matter emissions from sources other than engine exhaust gases is especially high in cities (i.e., in densely populated areas), where many people are exposed to emissions. Furthermore, the certain types of vehicle in city traffic can result in high-PM emissions from sources other than vehicle engines because of the need for braking and increasing velocity in an alternating sequence.

In view of these considerations, a practical conclusion can be drawn. While emissions of pollutants from engine exhaust gases are considerably limited by type-approval regulations, there are no limits in regards to particulate matter emissions from sources other than internal combustion engines. For road transport, PM emissions from vehicle engines are exclusively included in the emission inventory, and the concentrations of PM dimension fractions (and other substances) in the air as well as their effects are monitored. It is, therefore, advisable to undertake further study towards the control and reduction of PM road transport emissions from sources other than engine exhaust gases.

As part of the research, the authors plan to extend the scope to studies of the concentrations of individual dust fractions in places characteristic of road traffic. In this regard, in 2020, a doctoral dissertation was defended [46]. In this dissertation, a research methodology was developed that turned out to be effective. Therefore, it is advisable to implement this methodology on a larger scale, primarily in the field of systematic analysis of the results of empirical research at air quality monitoring stations.

**Author Contributions:** Conceptualization, K.B. and Z.C.; methodology, K.B., Z.C., K.S. and M.Z.-L.; Software, K.B. and Z.C.; Validation, K.B., Z.C., K.S. and M.Z.-L.; Formal analysis, K.B., Z.C. and H.S.; Investigation, K.B., Z.C., K.S. and M.Z.-L.; Resources, K.B., Z.C., K.S. and M.Z.-L.; Data curation, K.B., Z.C., H.S. and M.Z.-L.; Writing—original draft preparation, Z.C. and H.S.; Writing—review and editing, K.B., Z.C. and H.S.; Visualization, Z.C. and H.S.; Supervision, Z.C. and K.S.; Project administration, K.S.; Funding acquisition, K.S. All authors have read and agreed to the published version of the manuscript.

**Funding:** The APC was funded by the Institute of Environmental Protection—National Research Institute.

**Institutional Review Board Statement:** Not applicable.

**Informed Consent Statement:** Not applicable.

**Data Availability Statement:** Not applicable.

**Conflicts of Interest:** The authors declare no conflict of interest.

## Abbreviations

| | |
|---|---|
| BC | black carbon (soot) |
| $b_{PM}$ | specific distance particulate matter emission |
| $b_{PN}$ | specific distance number of particulate matter |
| C | coaches |
| E | national annual emission |
| EEA | European Environmental Agency |
| EMEP | European Monitoring and Evaluation Programme |
| $e_{PM}$ | specific brake emission of particulate matter |
| $e_{PN}$ | specific brake number of particulate matter |
| Er | relative national annual emission |
| ETC | European Transient Cycle |
| H | highway |
| HDT | heavy duty trucks |
| L | work |
| LCV | light commercial vehicles |

| | |
|---|---|
| LPG | liquefied petroleum gas |
| Mc | motorcycles |
| Mp | mopeds |
| $m_{PM}$ | emission of particulate matter |
| N | number of vehicles |
| P | annual mileage of vehicles |
| PC | passenger cars |
| PM10 | particulate matter with a mean aerodynamic diameter less than 10 μm |
| PM10-N-Exh | PM10—non-exhaust |
| PM10-T-PM10 | total |
| PM2.5 | PM with an average aerodynamic diameter less than 2.5 μm |
| PM2.5-N-Exh | PM2.5—non-exhaust |
| PM2.5-T | PM2.5—total |
| PM-TSP-T | TSP—total |
| PN | number of particulate matter |
| R | rural |
| s | distance |
| TSP | total suspended particles |
| TSP-N-Exh-TSP | non-exhaust |
| u | share of vehicle distance in traffic conditions in all distance |
| U | urban |
| UB | urban buses |
| V | vehicles |
| $v_{AV}$ | average velocity of vehicle |

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
