# Peer review of "Assessment of Environmental Risks of Particulate Matter Emissions from Road Transport Based on the Emission Inventory"

_applsci, doi:10.3390/app11136123_

Round 1

Reviewer 1 Report

This manuscript describes an original and interesting work. As such, it has the potential to be published in Applied Sciences. However, I have the following comments that the authors should implement in the revised manuscript before publication.

1) I suggest to merge Sections 1 and 2.

2) Resolution for Figures 6 and 7 should be improved.

3) Are the computational results grid-independent? The authors should comment on this key issue.

4) In the discussion, the practical impact of the results obtained in this work should be better highlighted. This should also be done in the section “Conclusions”.

5) Conclusions - The authors should give an outlook on future research work.

Author Response

Responses to the Reviewer No. 1

Dear Reviewer!

We would like to thank You very much for thorough acknowledging with our manuscript. Below we presented the answers for Your remarks.

Yours faithfully,

Authors.

Remark No. 1 of the Reviewer No. 1

1) I suggest to merge Sections 1 and 2.

Answer for Remark No. 1 of the Reviewer No. 1

Many publishing houses even require that the research methodology used in the article be separated in the form of a separate section, separate from the research results. This was also done in the article. However, sections 1.2 and 1.3 were combined.

Remark No. 2 of the Reviewer No. 1

2) Resolution for Figures 6 and 7 should be improved.

Answer for Remark No. 2 of the Reviewer No. 1

The resolution of drawings has been improved.

Remark No. 3 of the Reviewer No. 1

3) Are the computational results grid-independent? The authors should comment on this key issue.

Answer for Remark No. 3 of the Reviewer No. 1

All basic categories of motor vehicles are considered in the inventory. This is described in the section on the methodology used.

Remark No. 4 of the Reviewer No. 1

4) In the discussion, the practical impact of the results obtained in this work should be better highlighted. This should also be done in the section “Conclusions”.

Answer for Remark No. 4 of the Reviewer No. 1

In the discussion there has been indicated the practical aspect of the research results presented in the article.

Remark No. 5 of the Reviewer No. 1

5) Conclusions - The authors should give an outlook on future research work.

Answer for Remark No. 5 of the Reviewer No. 1

In the conclusions there has been suggested a continuation of the research, the results of which are presented in the article.

Reviewer 2 Report

The paper, after minor revision according to the attached report, deserve to be published. Best regards.

Author Response

Responses to the Reviewer No. 2

Dear Reviewer!

We would like to thank You very much for in-depth acknowledging with our manuscript. Below we presented the answers for Your remarks.

Yours faithfully,

Authors.

Remark No. 1 of the Reviewer No. 2

1) pg.1 lines 15 and 16, the Abstract is too long in my opinion, so I suggest to delete some phrases, at least ”Road transport is a source of particulate matter emissions that pose a serious threat to living organisms and the environment”;

Answer for Remark No. 1 of the Reviewer No. 2

The summary has been shortened.

Remark No. 2 of the Reviewer No. 2

2) pg.1 line 30, for a better diffusion of the paper I suggest to add ”PM10, PM25”;

Answer for Remark No. 2 of the Reviewer No. 2

Markings have been introduced.

Remark No. 3 of the Reviewer No. 2

3) pg.2 line 45, if the authors agree, I suggest to write ”There are distinguished [5-8, X, Y, 16-21]:”

Answer for Remark No. 3 of the Reviewer No. 2

As the Reviewer suggested, below presented publications have been cited in the revised version of the manuscript. Of course, we had to change the identity numbers of publications, because the order of them has been changed after revision.

[X – (We suppose that the Reviewer meant [X] instead of [4]). Cuspilici, A., Monforte, P., Ragusa, M.A.: Study of Saharan dust influence on PM10 measures in Sicily from 2013 to 2015. Ecological Indicators Vol. 76, 297–303, (2017). https://doi.org/10.1016/j.ecolind.2017.01.016.;

[Y] Pinto, J.T.D, Mistage, O., Bilotta, P., Helmers, E.: Road-rail intermodal freight transport as a strategy for climate change mitigation, Enviromental Development 25, 100–110, (2018).;

Remark No. 4 of the Reviewer No. 2

4) pg.5 line 179, I suggest to better explain why the progress is clearly visible;

Answer for Remark No. 4 of the Reviewer No. 2

An explanation has been added in the revised manuscript.

Remark No. 5 of the Reviewer No. 2

4) pg.14 line 457, it could be useful to write ”Then, from the above describes study it is possible to deduce that the average values of the relative national annual emission”;

Answer for Remark No. 5 of the Reviewer No. 2

A change was introduced in the revised manuscript.

Remark No. 6 of the Reviewer No. 2

6) pg.15 line 509, I think is more opportune to insert the Abbreviations at the beginning of the paper or, at least, announce them in the beginning. I think that put them in page 15 is not so clear for the readers.

Answer for Remark No. 6 of the Reviewer No. 2

The authors can of course do it, but they would prefer to leave it to the technical editorial office of the Publishing House, because publishers have different requirements in this regard.

Reviewer 3 Report

Abstract

The authors did not write what the purpose of the research was, they only wrote what they did. This should be completed as the article cannot be verified without knowing the purpose. The authors did not write what methodology they used. This needs to be completed.

Literature

There are no items in the bibliography in the form of articles by foreign authors (I do not include reports here) - only about 30% are articles by world authors - this is not enough. The literature review should be deepened with theories. Introduction You need to write a real introduction (according to common standards) - in this article the introduction is a review of reports and literature. The introduction should explain the research premises, provide research questions, thesis or hypothesis, and the purpose of the research. The structure / algorithm of the article should also be described. It is still unknown what the purpose of the research is and what the authors want to show, what research gap they want to fill.

Research

The authors presented the data as a study. No research method was used here, just a simple presentation. It is not enough - no idea. They basically present obvious things - where is the news here? Even when it comes to data presentation, there must be a purpose - and this purpose is unknown to us. I propose to rethink the concept of the article.

Conclusions

There is nothing new here. 

Author Response

Responses to the Reviewer No. 3

Dear Reviewer!

We would like to thank You very much for Your very important and priceless remarks regarding our manuscript.

Yours faithfully,

Authors.

Remark No. 1 of the Reviewer No. 3

Abstract

The authors did not write what the purpose of the research was, they only wrote what they did. This should be completed as the article cannot be verified without knowing the purpose. The authors did not write what methodology they used. This needs to be completed.

Answer for Remark No. 1 of the Reviewer No. 3

Corrections have been made.

Remark No. 2 of the Reviewer No. 3

Literature

There are no items in the bibliography in the form of articles by foreign authors (I do not include reports here) - only about 30% are articles by world authors - this is not enough. The literature review should be deepened with theories. Introduction You need to write a real introduction (according to common standards) - in this article the introduction is a review of reports and literature. The introduction should explain the research premises, provide research questions, thesis or hypothesis, and the purpose of the research. The structure / algorithm of the article should also be described. It is still unknown what the purpose of the research is and what the authors want to show, what research gap they want to fill.

Answer for Remark No. 2 of the Reviewer No. 3

There are representative items in the bibliography, showing, inter alia, mathematical models, including:

ChÅ‚opek, Z.; Suchocka, K.; Dudek, M.; Jakubowski, A. Hazards Posed by Polycyclic Aromatic Hydrocarbons Contained in the Dusts Emitted from Motor Vehicle Braking Systems. Archives of Environmental Protection 2016, Vol. 42 (3), 3–10. https://doi.org/10.1515/aep-2016-0033.

European Union emissions inventory report 2017 — European Environment Agency. Available online: https://www.eea.europa.eu/publications/european-union-emissions-inventory-report-2017 (accessed on 26.02.2021).

Karagulian, F.; Belis, C. A.; Dora, C. F. C.; Prüss-Ustün, A. M.; Bonjour, S.; Adair-Rohani, H.; Amann, M. Contributions to Cities’ Ambient Particulate Matter (PM): A Systematic Review of Local Source Contributions at Global Level. Atmos. Environ. 2015, 120, 475–483. https://doi.org/10.1016/j.atmosenv.2015.08.087.

Worldwide emission standards. Heavy duty & off-road vehicles. Delphi. Innovation for the real world. 2016/2017. Available online: https://www.delphi.com/sites/default/files/inline-files/2016-2017-heavy-duty-amp-off-highway-vehicles_0.pdf?sfvrsn=0.03636262961639791&status=Temp (accessed on 26.02.2021).

Panko, J. M.; Hitchcock, K. M.; Fuller, G. W.; Green, D. Evaluation of Tire Wear Contribution to PM2.5 in Urban Envi-ronments. Atmosphere 2019, 10 (2), 99. https://doi.org/10.3390/atmos10020099.

There are also source items with original research results, eg.:

Bebkiewicz, K.; ChÅ‚opek, Z.; Lasocki, J.; SzczepaÅ„ski, K.; Zimakowska-Laskowska, M. Inventory of Pollutant Emission from Motor Vehicles in Poland Using the COPERT 5 Software. Combust. Engines 2019, 178 (3), 150–154. https://doi.org/10.19206/CE-2019-326.

Bebkiewicz, K.; ChÅ‚opek, Z.; SzczepaÅ„ski, K.; Zimakowska-Laskowska, M. Results of Air Emission Inventory from Road Transport in Poland in 2014. Proc. Inst. Veh. 2017, 110, 77–88.

Lohmeyer A., Düring I.: Validierung von PM10-Immissionsberechnungen im Nahbereich von Straßen und Quantifizierung der Staubbildung von Straßen, Lützner Straße in Leipzig. Auftraggeber: Sächsisches Landesamt für Umwelt und Geologie, Dresden über Staatliche Umweltbetriebsgesellschaft, Radebeul, Februar 2001.

Penkała, M.; Ogrodnik, P.; Rogula-Kozłowska, W. Particulate Matter from the Road Surface Abrasion as a Problem of Non-Exhaust Emission Control. Environments 2018, 5 (1), 9. https://doi.org/10.3390/environments5010009.

Literature has been supplemented.

Remark No. 3 of the Reviewer No. 3

Research

The authors presented the data as a study. No research method was used here, just a simple presentation. It is not enough - no idea. They basically present obvious things - where is the news here? Even when it comes to data presentation, there must be a purpose - and this purpose is unknown to us. I propose to rethink the concept of the article.

Answer for Remark No. 3 of the Reviewer No. 3

Previously introduced changes clarify these ambiguities. In addition, the article presents the results of the analysis of the results of the inventory of pollutant emissions, which are not included in any of the world literature, e.g. the impact of the traffic model on the relative emission of particular fractions of solid particles.

Remark No. 4 of the Reviewer No. 3

Conclusions

There is nothing new here. 

Answer for Remark No. 4 of the Reviewer No. 3

The summary has been supplemented.

Round 2

Reviewer 3 Report

The authors corrected the article well, taking into account all comments. The article is better now. I appreciate the tenacity of the authors. Therefore, I recommend it for publication.